# Relationships Between Functional Impairment, Depressive Symptoms, and Ageing Attitudes in Older Adults

**DOI:** 10.3390/diagnostics15091145

**Published:** 2025-04-30

**Authors:** Jessica Sawang, Katya Numbers, Ben C. P. Lam, Simone Reppermund

**Affiliations:** 1Centre for Healthy Brain Ageing, Discipline of Psychiatry & Mental Health, Faculty of Medicine & Health, University of New South Wales (UNSW), Sydney 2052, Australia; 2School of Psychology and Public Health, La Trobe University, Melbourne 3086, Australia

**Keywords:** ageing attitudes, ageing, activities of daily living, IADL, depression, dementia risk

## Abstract

**Background/Objectives**: Negative attitudes towards ageing, depressive symptoms, and impairment in instrumental activities of daily living (IADL) are associated with worse health outcomes in older adults, including increased risk of dementia. Little is known about the longitudinal impact of depressive symptoms and functional impairment on ageing attitudes in older people. Identifying the relationships between these risk factors may help inform interventions targeting early dementia. The aim of this study was to determine whether depressive symptoms and functional impairment are associated with ageing attitudes over 6 years. **Methods**: Participants included 172 community-dwelling adults aged 76–96 years without dementia from the Sydney Memory and Ageing Study who were followed up over 6 years. Multiple linear regression models were used to examine prospective relationships between depressive symptoms, IADL (informant-reported or performance-based) and ageing attitudes. **Results**: After adjusting for potential confounding variables, more baseline depressive symptoms were associated with more negative ageing attitudes towards physical change (B = −0.10, 95%CI −0.18 to −0.02, *p* = 0.021) and psychological growth (B = −0.09, 95%CI −0.17 to −0.01, *p* = 0.037), and worse informant-reported IADL was associated with more negative ageing attitudes towards psychosocial change (B = −0.36, 95%CI −0.64 to −0.09, *p* = 0.010) over 6 years. **Conclusions**: The results highlight the importance of assessing and treating depressive symptoms and functional decline in older people, as they are significantly associated with negative attitudes about the ageing process, a known risk factor of worse health outcomes, including dementia.

## 1. Introduction

The global prevalence of dementia is projected to exceed 152 million by 2050 [1,2], making it the seventh leading cause of death across all age groups and the fourth leading cause for individuals aged 70 or older [3]. Approximately 40% of dementia cases are attributed to modifiable lifestyle factors, such as diet, physical activity, and mental health [4]. Consequently, understanding and mitigating these risk factors has become a global healthcare priority. However, while many studies focus on dementia-related risk factors, less attention has been paid to how psychological and functional factors are associated with attitudes toward ageing (ATA), which themselves are important predictors of health and well-being [5,6]. This is all the more important given the high prevalence of mental health conditions in an ageing population, with depression affecting approximately 35.1% of older adults worldwide [7] and the number of people aged over 60 expected to reach 2.1 billion by 2050 [8].

Attitudes toward ageing (ATAs) refer to an individual’s beliefs, perceptions, and experiences associated with the ageing process. These attitudes are multidimensional, encompassing both positive and negative dimensions [9], and are distinct from other constructs, such as depressive symptoms or overall well-being [10]. Positive ATA has been consistently linked with better physical health [2,5,6], cognitive performance, and longevity, as well as healthier behaviours, such as increased physical activity and adherence to medical advice [1]. Conversely, negative ATA is associated with poorer functional health, cognitive decline, and increased risk of mortality [11]. Importantly, research has shown that ATA not only predicts health outcomes but is also affected by psychological and functional factors over time [12,13,14].

Depressive symptoms are a significant psychological factor that may shape ATA [15,16,17,18,19]. Evidence suggests that depressive symptoms may affect ATA through mechanisms such as reduced self-efficacy, heightened perceptions of physical decline, and lower engagement in positive health behaviours [10]. These pathways highlight the complex interplay between mood and attitudes toward ageing, warranting further investigation. Depression is associated with a 50 per cent higher risk of developing dementia, and its severity has been shown to predict worse health outcomes [15]. Furthermore, there is evidence that depressive symptoms contribute to more negative ATA over time and that positive ATA may mitigate the progression of depressive symptoms [16,17]. However, the longitudinal interplay between depressive symptoms and ATA remains underexplored, particularly in older adults.

Functional impairment, particularly in instrumental activities of daily living (IADLs), is another factor that may affect ATA. IADLs encompass complex, everyday tasks such as managing finances, handling medications, and shopping, which require the integration of multiple cognitive and physical abilities [20]. Impairment in IADL has been identified as a key marker of early cognitive decline and a predictor of dementia risk [21,22,23]. IADLs are commonly assessed using informant-reported measures, such as the Bayer-Activities of Daily Living (B-ADL) scale [24], or performance-based tasks, such as the Sydney Test of Activities of Daily Living in Memory Disorders (STAM) [25]. Recent studies from McGarrigle et al. (2022) and Diehl et al. (2020) suggest that performance-based assessments may offer greater sensitivity in detecting early functional decline, while informant-reported measures can capture a broader context of daily challenges [6,26]. Both approaches underscore the relationship between functional capacity and ATA, providing complementary insights. While research indicates that positive ATA is associated with better future IADL [26,27] and negative ATA with greater IADL impairment [26], little is known about how different measures of IADL—such as informant-reported versus performance-based assessments—might differentially predict ATA over time. Given that performance-based IADL measures are less affected by biases such as mood or carer burden [25], they may offer unique insights into the relationship between functional decline and ATA.

Despite substantial research on ATA as predictors of health outcomes, fewer studies have examined the reverse relationship: how psychological and functional factors, such as depressive symptoms and IADL, are longitudinally associated with ATA. Understanding these relationships is critical, as ATAs are modifiable and have the potential to mediate health outcomes in older adults [1]. Interventions such as cognitive-behavioural therapy to improve ageing attitudes or programs targeting functional independence through physical rehabilitation and support with IADL have shown promise in promoting positive ATA [5,6]. These practical applications highlight the importance of addressing both psychological and functional domains to foster healthier ageing trajectories. Moreover, the distinction between informant-reported and performance-based IADL assessments may provide nuanced insights that are relevant for designing targeted interventions for improving ATA.

Furthermore, despite growing evidence linking depressive symptoms and functional impairment to more negative attitudes towards ageing (ATA), key gaps remain. Most prior research has relied on self-reported functional measures and examined global ATA rather than specific domains (e.g., psychosocial loss, physical change, psychological growth). Additionally, the relative contributions of performance-based versus informant-reported functional measures to the development of ageing attitudes have not been compared. This is particularly important in very old adults (aged 75 and over), a population at increased risk for functional decline, depressive symptoms, and internalised ageism [28,29] but often underrepresented in longitudinal research. The current study addresses these gaps by examining how depressive symptoms and both informant-reported and performance-based instrumental activities of daily living (IADLs) predict three distinct domains of ageing attitudes (physical change, psychosocial loss, and psychological growth) over a six-year period in a large cohort of community-dwelling older adults, aged 75 and above. Few studies have explored these relationships longitudinally or in very old adults—a group particularly vulnerable to both internalised stigma and adverse health outcomes. Drawing on stereotype embodiment theory [30], which posits that internalised ageing stereotypes impact health trajectories, we explored how psychological and functional experiences shape older adults’ perceptions of ageing. By examining these relationships, this research contributes to a growing body of literature on ageing attitudes and their role in promoting healthy ageing [5]. In doing so, the findings from this study can inform the development of interventions targeting depressive symptoms and functional impairments to improve attitudes toward ageing and, ultimately, health outcomes in older populations.

## 2. Materials and Methods

### 2.1. Participants

This study employed a longitudinal observational design to examine predictors of ageing attitudes over a six-year period in a sample of 1037 community-dwelling individuals aged 70–90 years without dementia at baseline from the Sydney Memory and Ageing Study (MAS) [31]. Participants were randomly selected through the Electoral Roll in Sydney’s East and were required to be proficient enough in English to complete a psychometric assessment and consent to participate in this study. Participants with psychotic symptoms, schizophrenia or bipolar disorder, developmental disabilities, multiple sclerosis, or other conditions that potentially could have prevented them from completing assessments were excluded. Recruitment and initial assessment occurred between 2005 and 2007 [31]. Participant assessments were conducted every two years (called a ‘wave’) by trained research assistants at either a study centre or participants’ own homes. Assessments at each follow-up involved an interview, questionnaires, a medical examination, and a comprehensive neuropsychological assessment. Informants, who were close friends or family of participants who knew them well enough to report on their functional status and spent at least one hour per week with the participant, completed questionnaires and a phone interview about the participant.

For this study, the baseline was wave 4; as this was the first wave, the performance-based IADL measure was introduced, and participants were followed over 6 years until wave 7. Wave 4 included 708 participants. Participants with a previous or new diagnosis of dementia by wave 4 were excluded (n = 82). Further, participants were only included in the present study if they remained in the study at wave 7 and had complete ATA data. Thus, for the present study, 172 participants with complete wave 7 ATA data were included. Of these, the majority had a study informant at wave 4 (n = 149, 86.6%). Participants’ mean age was 81.2 years (SD = 3.6), 60.5% were women (n = 104), 98.8% were Caucasian (n = 170), and the mean years of education was 12.5 (SD = 3.4). Figure 1 presents a flowchart of the sample selection process for this study.

Participants and informants gave informed consent. This study was approved by the Human Research Ethics Committee of the University of New South Wales (HC: 05037, 09382, 14327, 190962).

### 2.2. Measures

Attitudes towards ageing (ATA) were measured using the self-rated 12-item Attitudes to Ageing Questionnaire, Short Form (AAQ-SF) [9,32], a psychometrically robust measure of ATA [9]. The AAQ-SF is comprised of 3 subscales with 4 items each. The Psychological Growth subscale assessed attitudes towards perceived gains associated with age; the Physical Change subscale focused on perceived health and vitality in relation to age; the Psychosocial Loss subscale reflected the apparent psychological and social losses experienced as participants age. Participants were asked to rate their feelings according to statements (e.g., “It is a privilege to grow old”, “My health is better than I expected for my age”) on a 5-point scale (1 “Strongly Disagree” to 5 “Strongly Agree”) [9]. The Psychosocial Loss subscale was reverse scored as it had negatively worded statements and thereafter labelled as Psychosocial Change. AAQ-SF subscale scores were averaged across their four items. For AAQ-SF scores, higher scores indicate more positive ATA. Imputation was used to impute missing item scores, and Confirmatory Factor Analysis (CFA) was conducted to examine the factor structure (see Statistical Analysis for details).

Depressive symptoms were assessed using the self-reported, 15-item Geriatric Depression Scale (GDS-15) [33,34], which has been shown to be a valid and reliable indicator of depressive symptoms in older adults [9]. Participants answered “yes” or “no” questions with total scores ranging from 0 to 15, where a higher score indicates more depressive symptoms. In the MAS study, the GDS-15 version with item 9, as described by Brink [35], was used. Scores were pro-rated to account for missing data.

IADL were assessed through two measures: the informant-reported Bayer-Activities of Daily Living (B-ADL) [24] scale and the performance-based Sydney Test of Activities of Daily Living in Memory Disorders (STAM) [25]. The B-ADL [24] is a 25-item questionnaire that has good validity and reliability in measuring IADL impairment [36]. The B-ADL was completed by the participant’s informant. Each question asked how often the participant has difficulties completing a specific activity (e.g., using a telephone, shopping, counting out money, etc.) and was scored on a 10-point scale, with 1 indicating “Never” and 10 indicating “Always”. Scores were pro-rated to account for missing data. Higher scores indicate greater IADL impairment. The Sydney Test of Activities of Daily Living in Memory Disorders (STAM) [25] is a performance-based measure of IADL that consists of 9 tasks covering the functional domains of communication, dressing, handling finances, managing everyday activities, time orientation, medication management, shopping, counting money, and memory. Each of these 9 tasks contains 4 items, resulting in a maximum total score of 36. Higher scores indicate better IADL function. STAM scores were adjusted in 3 ways: items were scored as missing if a physical disability prevented a participant from performing a task, scores were adjusted and penalised if the time taken to complete a task was over the time limit, and the total score was adjusted for the maximum possible score of valid items, which excluded missing items. The STAM has been shown to be a valid and reliable measure of IADL performance in those with normal cognition, MCI, and dementia [25].

Covariates included participants’ age, sex, and years of education, as well as global cognition, cardiovascular risk, vision impairment and arthritis. These covariates were included as they have been shown to impact IADL performance [25,37,38] and may have an effect on ATA. Global cognition was assessed through a comprehensive neuropsychological test battery measuring memory, attention, language, visuospatial ability, and executive function [31]. Global cognition composite scores (interpreted as z-scores) were computed by averaging the domain scores and standardising them against a baseline healthy subsample, with higher scores indicating better cognitive performance [39]. Cardiovascular risk (CVR) scores were calculated according to the Framingham Stroke Study [40], which incorporates diabetic status, total cholesterol level, high-density lipoprotein level, systolic blood pressure, anti-hypertensive treatment, and smoking status. Vision impairment and arthritis status were self-reported at baseline, and participants answered if, in the last two years, they had any visual impairment and if they were diagnosed with arthritis.

### 2.3. Statistical Analysis

Baseline demographics and other relevant covariates, including global cognition, CVR scores, arthritis, and vision impairment, were compared between the completers and non-completers of the AAQ-SF at wave 7. Demographics and covariates at baseline (wave 4) for the 172 participants included in the current study and the 537 excluded participants were compared to examine whether there were group-level differences at baseline between completers and non-completers. Baseline demographics and covariates were also compared between the AAQ-SF completers who were included and excluded from the multiple linear regressions.

Across the 12-item AAQ-SF, missing data were between 1.2–5.2%. Moreover, among the 26 participants with missing data, the majority (n = 23) had three or fewer items missing. Given that there were relatively few missing data points and that pro-rating across subscales was not appropriate, missing AAQ-SF item scores were imputed. Similar mean and distribution of AAQ-SF scores were obtained using the raw and imputed data.

CFA using the maximum likelihood estimation was conducted to examine the proposed three-factor structure. The model fitted the data adequately after specifying two additional residual covariances: χ2(49) = 100.084, *p* < 0.001, CFI = 0.914, RMSEA = 0.078, and SRMR = 0.066. All items significantly loaded onto their corresponding factors. Factor correlations ranged from 0.239 to 0.369. These results support the notion that the AAQ-SF subscales need to be examined separately [9].

The relationships between AAQ-SF, GDS-15, B-ADL, and STAM scores at wave 4 and wave 7 were investigated through Spearman’s rank order correlation, given that some of these variables were skewed.

Multiple linear regressions were conducted to assess the prospective relationships between the GDS-15, B-ADL, and STAM at wave 4 and AAQ-SF subscale scores at wave 7, controlling for age, sex, years of education, global cognition, CVR scores, arthritis, and vision impairment. Given that these analyses involving the three AAQ-SF subscales are exploratory in nature, *p*-value adjustment was not implemented. The level of significance was set to *p* < 0.05 (two-sided).

All analyses were conducted in IBM SPSS Statistics 28. Prior to regression modelling, assumptions for linear regression were evaluated, including normality of residuals, linearity, homoscedasticity, and multicollinearity. Variance Inflation Factors (VIFs) were computed for all predictors and found to be below 2.0.

Covariates were selected based on the prior literature and theoretical models indicating their influence on both ageing attitudes and functional ability, including age, sex, years of education, global cognition (ACE-III), cardiovascular disease burden, arthritis, and visual impairment [41].

This article was written using Microsoft Word Office 2024.

## 3. Results

At wave 4 (baseline), there were 708 participants; 6 years later, at wave 7, 537 were lost to follow-up. Compared with the present study’s sample, participants lost to follow-up were older, had fewer years of education, worse cognition, greater incidence of visual impairment, more depressive symptoms, and worse IADL at baseline (see Appendix A Table A1). Of the 315 participants included in wave 7, 143 were not administered the AAQ-SF questionnaire. Compared with those who completed the AAQ-SF, participants who did not complete the AAQ-SF were older, had fewer years of education, worse cognition, more depressive symptoms, and more functional impairment at baseline (see Appendix A Table A2). Of the 172 participants who completed the AAQ-SF, 20 participants were not included in the GDS-15 linear regressions, 37 participants were not included in the B-ADL linear regressions, and 20 participants were not included in the STAM linear regressions. Compared to those included in the linear regressions, participants who were excluded from those analyses did not have any significant demographic differences at baseline (see Appendix A Table A3, Table A4 and Table A5).

For the Physical Change AAQ-SF subscale, participants had a mean score of 3.60, an SD of 0.81, and a range of 1.25–5.00. For the Psychosocial Change AAQ-SF subscale, participants had a mean score of 3.23, an SD of 1.00, and a range of 1.00–5.00. For the Psychological Growth AAQ-SF subscale, participants had a mean score of 4.00, an SD of 0.83, and a range of 1.00–5.00.

Table 1 displays Spearman correlations between AAQ-SF scores at wave 7 and GDS-15, B-ADL, and STAM scores at wave 4 and wave 7. Physical Change AAQ-SF subscale scores were significantly and negatively correlated with GDS-15 (ρ = −0.282, *p* < 0.001) at wave 4 and wave 7 (ρ = −0.155, *p* = 0.049), and significantly, positively correlated with the STAM (ρ = 0.229, *p* = 0.014) at wave 7. Psychosocial Change AAQ-SF subscale scores were significantly and negatively correlated with GDS-15 (ρ = −0.189, *p* = 0.014) at wave 4. Psychological Growth AAQ-SF subscale scores were significantly and negatively associated with GDS-15 (ρ = −0.152, *p* = 0.049) at wave 4.

Table 2 displays the regression results examining the relationship between baseline GDS-15 and wave 7 Physical Change AAQ-SF subscale scores, controlling for covariates in Step 1. After adjusting for all covariates, GDS-15 significantly predicted Physical Change AAQ-SF subscale scores (B = −0.10, 95%CI −0.18 to −0.02, *p* = 0.021), accounting for 3.3% of additional variance. Wave 7 Physical Change AAQ-SF subscale scores were not significantly associated with B-ADL (B = −0.19, 95%CI −0.42 to 0.03, *p* = 0.091) and STAM scores at baseline (B = 0.00, 95%CI −0.04 to 0.04, *p* = 0.946). Detailed results are summarised in the Appendix A Table A6 and Table A7.

Table 3 displays the regression results examining the relationship between baseline B-ADL and wave 7 Psychosocial Change AAQ-SF subscale scores, controlling for covariates at Step 1. After controlling for all covariates, B-ADL significantly predicted Psychosocial Change AAQ-SF subscale scores (B = −0.36, 95%CI −0.64 to −0.09, *p* = 0.010), accounting for 4.7% of the additional variance. Wave 7 Psychosocial Change AAQ-SF subscale scores were not significantly associated with GDS-15 (B = −0.10, 95%CI −0.21 to 0.01, *p* = 0.071) and STAM scores at baseline (B = −0.04, 95%CI −0.09 to 0.01, *p* = 0.139). Detailed results are summarised in the Appendix A Table A8 and Table A9.

Table 4 displays the regression results examining the relationship between baseline GDS-15 and wave 7 Psychological Growth AAQ-SF subscale scores, controlling for covariates at Step 1. After controlling for all covariates, GDS-15 significantly predicted Psychological Growth AAQ-SF subscale scores (B = −0.09, 95%CI −0.17 to −0.01, *p* = 0.037), accounting for 2.8% of additional variance. Wave 7 Psychological Growth AAQ-SF subscale scores were not significantly associated with B-ADL (B = −0.10, 95%CI −0.30 to 0.10, *p* = 0.340) and STAM scores at baseline (B = −0.03, 95%CI −0.07 to 0.01, *p* = 0.113). Detailed results are summarised in the Appendix A Table A10 and Table A11.

## 4. Discussion

This study aimed to examine the cross-sectional and prospective associations between depressive symptoms, informant-reported and performance-based IADL, and ATA 6 years later in a community-dwelling sample of older adults.

The correlations between AAQ-SF subscales, depressive symptoms, and IADL (informant-report and performance-based) showed that depressive symptoms were associated with worse attitudes towards physical change cross-sectionally. These results are in line with the literature, which has also demonstrated that attitudes towards physical change are associated with depressive symptoms [12,42] and diagnosed depression [43]. One interpretation of these results may be that the Physical Change subscales of the AAQ-SF assess concepts similar to those captured by the GDS-15, such as energy levels. In contrast to previous research [12,42,43], this study found that attitudes towards psychosocial change and psychological growth were not significantly associated with depressive symptoms cross-sectionally. However, the three studies that found this relationship to be significant used the full 24-item AAQ to measure ATA, meaning the Psychosocial Change (i.e., Psychosocial Loss) and Psychological Growth subscales contained four additional questions each compared with the AAQ-SF. The additional questions in the full AAQ may better capture attitudes in both the Psychosocial Change and Psychological Growth subscale, and excluding these may explain the lack of cross-sectional associations with depression found in this study. Other explanations may be that studies where a significant association between Psychosocial Change and Psychological Growth and depression were found to have larger [12,42,43] and younger samples [43] compared with the present study.

This study also found that more negative attitudes towards physical change were associated with worse performance-based IADL cross-sectionally. The Physical Change subscale asks about things that would be affected by functional difficulties, such as exercising, perceptions of health, energy, and feeling old. These functional difficulties may be more likely to be captured by a performance-based measure of IADL compared with the subjective informant-reported IADL measure. Finally, neither measure of IADL was associated with Psychosocial Change or Psychological Growth scores cross-sectionally. The lack of associations could be due to these subscales measuring concepts not directly affected by functional impairment. Previous research on the associations between IADL and AAQ scores did not examine AAQ subscales, nor did they examine the differences in these associations between performance-based and informant-reported IADL measures [11], meaning these results are novel and should be replicated in future studies with larger samples.

This study showed that more depressive symptoms at baseline were associated with more negative perceived physical change and less perceived psychological growth after 6 years. This finding is consistent with other studies that have shown associations between depression and negative ageing attitudes longitudinally [17,19]. Indeed, depressive symptoms may contribute to more negative perceptions of physical health and less positive perceptions of psychological growth, as well as ageing, as depression has been linked to worse quality of life both cross-sectionally and over time [44], and as an extension, depression could also affect perceptions of the ageing process. Baseline depressive symptoms were not significantly associated with more perceived psychosocial loss, which may have been due to the Psychosocial Change subscale containing items such as “I feel excluded from things because of my age”, which may be affected more by a person’s social climate and its inclusivity and attitudes towards older people, than a person’s depressive symptoms.

More informant-reported functional decline at baseline was associated with more negative self-perceptions of psychosocial change after 6 years. Interestingly, informant-reported functional decline was not significantly associated with self-perceptions of psychological growth. This suggests that psychosocial change and psychological growth, while theoretically related, may represent distinct concepts of ageing for older adults. Additionally, it was notable that informant-reported functional changes were not linked to self-reported physical changes despite many studies demonstrating a strong association between physical disability and poorer self-reported functional ability [45,46,47,48]. This discrepancy may be due to our control of key physical covariates (e.g., arthritis and vision impairment) known to impact IADL performance [37,38] in our analysis, or it may be due to using an informant-reported IADL measure, as informants may have less insight into subtle functional and physical changes that a self-reported measure may capture. While several studies have shown that IADL impairment is associated with negative ATA cross-sectionally [11,49], only one study by Kwak et al. [50] has investigated whether IADL is related to ATA longitudinally. Kwak et al. found that more hours of functional care, a proxy measure of IADL impairment, were related to more negative ATA over time. Further, our finding that informant-reported IADL was significantly predictive of psychosocial change, while depressive symptoms were not, is novel and suggests that while IADL and depression are related [51,52] and they predict each other over time [53,54], IADL impairment may be related to future ATA, or at least attitudes towards psychosocial change through a mechanism unrelated to mood.

A surprising finding from this study was that while self-reported GDS-15 and informant-reported IADL were related to the various AAQ-SF subscales, performance-based IADL scores were not. This could be because the AAQ-SF, the GDS-15, and the Bayer-IADL are all subjective measures, whereas the STAM is not. Further, studies have shown that informant-reported IADLs are known to be impacted by participants’ moods and personalities, while performance-based measures are not [55]. Another methodological explanation may be that at baseline, participants who completed the performance-based IADL were highly functioning and largely showed little to no objective functional impairment on the STAM. This is compared to a much larger range of scores reported by the informant on the B-ADL. As such, the non-significant result could be due to the limited variance of STAM scores at baseline.

These findings highlight the clinical and public health relevance of addressing ageing attitudes as modifiable risk factors in later life. Evidence suggests that negative attitudes toward ageing are linked to poorer psychological well-being, increased disability, and accelerated cognitive decline [1,11]. Our results indicate that depressive symptoms and informant-reported functional limitations—both potentially modifiable—are key predictors of less adaptive ageing attitudes. These findings support the value of early interventions targeting both psychological health and everyday function, using approaches such as cognitive behavioural therapy, social prescribing, or functional reablement to foster more positive self-perceptions of ageing [56,57].

Several limitations of the current study should be considered. Firstly, the original MAS sample is not representative of the general older Australian population. MAS participants were highly educated, community-dwelling, overwhelmingly Caucasian, and fluent in English. Secondly, the attrition during the present study’s 6-year follow-up was non-random. Participants lost to follow-up were older, had fewer years of education, and had worse psychological and physical functioning. Thirdly, ATA was the main outcome variable of this study but was only measured once, at wave 7, meaning that examining change in ATA over time was not possible. Further, the AAQ-SF was administered midway through wave 7 as an optional questionnaire sent to participants, meaning it is possible only the healthier participants completed it. Indeed, out of the already biased and healthier wave 7 participants, those who completed the AAQ-SF were younger, had more years of education, and were cognitively, psychologically, and physically healthier. Fourthly, depressive symptoms and ATA were assessed through self-reported measures and thus are susceptible to self-report bias; that is, participants may underreport or overreport depressive symptoms or ATA due to a range of factors, including cognitive decline [58]. Finally, the STAM tool—while a validated performance-based IADL assessment—may have been limited by ceiling effects in our high-functioning sample, potentially underestimating associations with ageing attitudes. This study also excluded participants who had dementia at baseline (wave 4), meaning they could not, by definition, have major impairments in IADL, which likely contributed further to the restricted range of STAM scores.

This study’s strengths include the relatively long follow-up period of 6 years. Other strengths include a well-characterised sample, comprehensive neuropsychological testing, medical history, questionnaires, and a medical exam, which were considered in consensus diagnoses made by a panel of experts. Further, this is the only study to examine the relationships between both a performance-based and informant-reported IADL measure, depression, and attitudes towards ageing. This study also included arthritis and vision impairment as potential confounding variables, which are not often controlled for, meaning that IADL function may be more representative of functional decline that is not due to physical disability.

## 5. Conclusions

This study shows that depressive symptoms were associated with worse future ageing attitudes towards physical change and psychological growth, while informant-based IADL was associated with future attitudes towards psychosocial change. Given that IADL, depressive symptoms, and ATA are related to the progression to dementia and that each of these factors is also modifiable, there is great merit in understanding the complex and potentially bi-directional relationships between these variables. Together, these results highlight the importance of assessing depressive symptoms and capturing informant-reported functional decline, as these are significantly associated with worse attitudes about the ageing process, a known risk factor of worse health outcomes, including dementia. A life course approach of interventions targeting depression and functional decline early may improve ageing attitudes and thus the health of older adults, therefore reducing the lifetime risk of dementia.

## Figures and Tables

**Figure 1 diagnostics-15-01145-f001:**
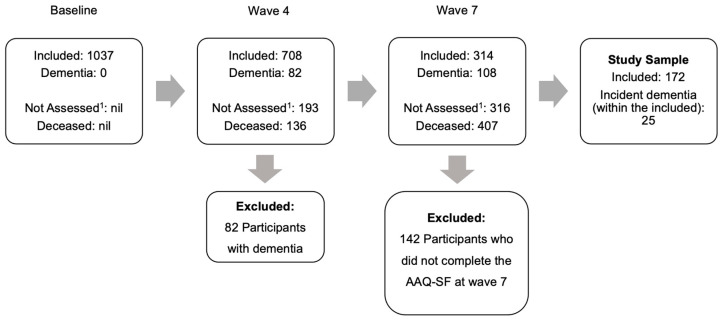
Flowchart of the inclusion and exclusion of participants in the present study. ^1^ This includes participants who were not assessed, had withdrawn, or were lost to follow-up.

**Table 1 diagnostics-15-01145-t001:** Spearman correlation between AAQ-SF, GDS, B-ADL, and STAM.

Variable	n	1	2	3	4	5	6	7	8	9
1. Physical Change AAQ-SF subscale score at wave 7	172	-								
2. Psychosocial Change AAQ-SF subscale score at wave 7	172	0.205 **	-							
3. Psychological Growth AAQ-SF subscale Score at wave 7	172	0.291 **	0.056	-						
4. GDS-15 at wave 4	169	−0.282 **	−0.189 *	−0.152 *	-					
5. B-ADL score at wave 4	149	−0.160	−0.147	−0.043	0.274 **	-				
6. STAM score at wave 4	159	0.108	0.032	−0.158	−0.051	−0.235 **	-			
7. GDS-15 score at wave 7	162	−0.155 *	−0.104	−0.058	0.249 **	−0.001	−0.132	-		
8. B-ADL score at wave 7	142	−0.119	−0.065	−0.041	0.239 **	0.639 **	−0.182 *	0.133	-	
9. STAM score at wave 7	115	0.229 *	0.048	0.006	−0.263 **	−0.254 *	0.440 **	−0.347 **	−0.370 **	-

AAQ-SF, Attitudes towards Ageing Questionnaire (Short Form); GDS-15, 15-item Geriatric Depression Scale; B-ADL, Bayer-Activities of Daily Living Scale; STAM, Sydney Test of Activities of Daily Living in Memory Disorders. * *p* < 0.05. ** *p* < 0.01.

**Table 2 diagnostics-15-01145-t002:** Linear regression of GDS-15 predicting Physical Change AAQ-SF subscale scores at wave 7 (n = 152).

	Variables	B	SE	95% CI for B		β	*p*	R^2^	ΔR^2^
				Lower	Upper				
Step 1							0.038	0.10	0.10
	Age	−0.04	0.02	−0.07	0.00	−0.15	0.075		
	Sex	−0.20	0.14	−0.47	0.07	−0.12	0.152		
	Education	0.04	0.02	−0.01	0.08	0.15	0.087		
	Global cognition	0.01	0.07	−0.13	0.15	0.02	0.841		
	CVR score	−0.03	0.02	−0.07	0.01	−0.12	0.142		
	Arthritis	−0.07	0.13	−0.34	0.19	−0.04	0.592		
	Vision impairment	−0.07	0.24	−0.55	0.42	−0.02	0.783		
Step 2							0.021	0.13	0.03
	GDS-15 score	−0.10	0.04	−0.18	−0.02	−0.19	0.021		

CVR, Cardiovascular Risk (Framingham); GDS-15, 15-item Geriatric Depression Scale.

**Table 3 diagnostics-15-01145-t003:** Linear regression of B-ADL predicting Psychosocial Change AAQ-SF subscale scores at wave 7 (n = 135).

	Variables	B	SE	95% CI for B		β	*p*	R^2^	ΔR^2^
				Lower	Upper				
Step 1							0.081	0.09	0.09
	Age	−0.02	0.03	−0.07	0.03	−0.06	0.494		
	Sex	0.60	0.19	0.23	0.97	0.29	0.002		
	Education	0.01	0.03	−0.04	0.06	0.04	0.703		
	Global cognition	0.04	0.10	−0.16	0.23	0.04	0.705		
	CVR score	−0.01	0.03	−0.06	0.04	−0.03	0.770		
	Arthritis	−0.16	0.18	−0.52	0.19	−0.08	0.362		
	Vision impairment	0.21	0.32	−0.42	0.85	0.06	0.505		
Step 2							0.010	0.14	0.05
	B-ADL score	−0.36	0.14	−0.64	−0.09	−0.23	0.010		

CVR, Cardiovascular Risk (Framingham); B-ADL, Bayer-Activities of Daily Living Scale.

**Table 4 diagnostics-15-01145-t004:** Linear regression of GDS-15 predicting Psychological Growth AAQ-SF subscale scores at wave 7 (n = 152).

	Variables	B	SE	95% CI for B		β	*p*	R^2^	ΔR^2^
				Lower	Upper				
Step 1							0.138	0.07	0.07
	Age	−0.01	0.02	−0.05	0.02	−0.06	0.460		
	Sex	−0.01	0.14	−0.29	0.26	−0.01	0.925		
	Education	−0.04	0.02	−0.08	−0.00	−0.19	0.039		
	Global cognition	0.03	0.07	−0.11	0.17	0.04	0.635		
	CVR score	−0.04	0.02	−0.08	−0.00	−0.18	0.035		
	Arthritis	−0.04	0.13	−0.31	0.22	−0.03	0.741		
	Vision impairment	0.37	0.24	−0.11	0.85	0.12	0.132		
Step 2							0.037	0.10	0.03
	GDS-15 score	−0.09	0.04	−0.17	−0.01	−0.18	0.037		

CVR, Cardiovascular Risk (Framingham); GDS-15, 15-item Geriatric Depression Scale.

## Data Availability

The datasets that support the findings of this study are not publicly available due to the terms of consent for research participation stipulating that an individual’s data can only be shared outside of the MAS investigators group if the group has reviewed and approved the proposed secondary use of the data. This consent applies regardless of whether data have been de-identified. Access to the datasets is mediated via a standardised request process managed by the CHeBA Research Bank, which can be contacted at ChebaData@unsw.edu.au.

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
