# Peer review of "Relationships Between Functional Impairment, Depressive Symptoms, and Ageing Attitudes in Older Adults"

_diagnostics, 2025, doi:10.3390/diagnostics15091145_

Round 1
Reviewer 1 Report
Comments and Suggestions for Authors
The article presents an interesting topic in relation to functional deterioration, depressive symptoms, and attitudes of aging in community-dwelling older people.
Major observations:
1. In the introduction, authors should more thoroughly justify the central theme of the article, especially in terms of its novelty. The current rationale lacks depth to contextualize the study within the existing literature and fails to clearly communicate contributions of this research. A more solid theoretical framework would significantly strengthen the manuscript, specifying what would be the relevance of delving into this topic.
2. In the methodology, although the study design is understood by reading the methods section, it should be explicitly described at the beginning of the first paragraph.
3. Regarding the statistical analysis, I suggest that the authors specify which assumptions of linear regression were evaluated and how this evaluation was performed. In particular, the multicollinearity analysis should be detailed. Additionally, a justification must be included on the criteria used to incorporate covariates into the adjusted model.
4. In the discussion, an expanded analysis of the clinical (or may be public health?) relevance of the findings is necessary. It is also necessary to explore the possible effects of limitations on the results, detailing how the research team mitigated such impacts and why the findings remain valid.
Minor observation:
1. Basically, it is necessary to explore what is the unanswered research question in the literature that the authors intend to answer or at least what gap within this problem they want to fill.
Author Response
Comment 1: The article presents an interesting topic in relation to functional deterioration, depressive symptoms, and attitudes of aging in community-dwelling older people.
Response 1: We thank the reviewer for their positive feedback and helpful suggestions.
Comment 2: Major observations: 1. In the introduction, authors should more thoroughly justify the central theme of the article, especially in terms of its novelty. The current rationale lacks depth to contextualize the study within the existing literature and fails to clearly communicate contributions of this research. A more solid theoretical framework would significantly strengthen the manuscript, specifying what would be the relevance of delving into this topic.
Response 2: We appreciate this feedback and have revised the introduction to better contextualise our study within ageing theory. Specifically, we now draw on Levy’s stereotype embodiment theory (Levy, 2009) to highlight how depressive symptoms and functional limitations may internalise negative ageing beliefs over time. We also clarify that our study is novel in its dual focus on performance-based and informant-reported IADL, and in its analysis of distinct domains of ageing attitudes (psychosocial loss, physical change, and psychological growth) over a six-year follow-up in very old adults.
(See Introduction, lines 98 - 115.)
“Furthermore, despite growing evidence linking depressive symptoms and functional impairment to more negative attitudes toward ageing (ATA), key gaps remain. Most prior research has relied on self-reported functional measures and examined global ATA rather than specific domains (e.g., psychosocial loss, physical change, psychological growth). Additionally, the relative contributions of performance-based versus informant-reported functional measures to the development of ageing attitudes have not been compared. This is particularly important in very old adults (aged 75 and over), a population at increased risk for functional decline, depressive symptoms, and internalised ageism [28, 29], but often underrepresented in longitudinal research. The current study addresses these gaps by examining how depressive symptoms and both informant-reported and performance-based instrumental activities of daily living (IADLs) predict three distinct domains of ageing attitudes (physical change, psychosocial loss, and psychological growth) over a six-year period in a large cohort of community-dwelling older adults, aged 75 and above. Few studies have explored these relationships longitudinally or in very old adults—a group particularly vulnerable to both internalised stigma and adverse health outcomes. Drawing on stereotype embodiment theory [30], which posits that internalised ageing stereotypes impact health trajectories, we explore how psychological and functional experiences shape older adults’ perceptions of ageing.”
Comment 3: 2. In the methodology, although the study design is understood by reading the methods section, it should be explicitly described at the beginning of the first paragraph.
Response 3: We agree and have added a sentence explicitly identifying the design as a longitudinal observational study at the start of the Participants section.
(See Materials and Methods, lines 122 - 125.)
“This study employed a longitudinal observational design to examine predictors of ageing attitudes over a six-year period in a sample of 1,037 community-dwelling individuals aged 70-90 years without dementia at baseline, from the Sydney Memory and Ageing Study (MAS) [31].”
Comment 4: 3. Regarding the statistical analysis, I suggest that the authors specify which assumptions of linear regression were evaluated and how this evaluation was performed. In particular, the multicollinearity analysis should be detailed. Additionally, a justification must be included on the criteria used to incorporate covariates into the adjusted model.
Response 4: We have revised the end of Statistical Analysis, to state that assumptions for linear regression were tested, including normality, linearity, homoscedasticity, and multicollinearity (assessed via Variance Inflation Factors, all <2). We also justify our covariate selection based on both theoretical models and previous findings linking these factors (e.g., cardiovascular burden, cognition, arthritis) to functional decline and ageing attitudes.
(See Materials and Methods, lines 239 - 246.)
“All analyses were conducted in IBM SPSS Statistics 28. Prior to regression modelling, assumptions for linear regression were evaluated, including normality of residuals, linearity, homoscedasticity, and multicollinearity. Variance Inflation Factors (VIFs) were computed for all predictors and found to be below 2.0.
Covariates were selected based on prior literature and theoretical models indicating their influence on both ageing attitudes and functional ability, including age, sex, years of education, global cognition (ACE-III), cardiovascular disease burden, arthritis, and visual impairment [41].”
Comment 5: 4. In the discussion, an expanded analysis of the clinical (or may be public health?) relevance of the findings is necessary. It is also necessary to explore the possible effects of limitations on the results, detailing how the research team mitigated such impacts and why the findings remain valid.
Response 5: We have added a new paragraph to the Discussion highlighting the clinical and public health relevance of our findings. Specifically, we note that ageing attitudes are modifiable, and may serve as valuable targets in interventions addressing depressive symptoms and functional decline. We reference potential strategies such as cognitive-behavioural therapy and functional reablement approaches that may foster more positive self-perceptions of ageing.
Additionally, we have expanded the Limitations section to note that the STAM tool may have shown reduced variability in this high-functioning cohort.
(See Discussion, lines 395–403 and 419 - 421.)
“These findings highlight the clinical and public health relevance of addressing ageing attitudes as modifiable risk factors in later life. Evidence suggests that negative attitudes toward ageing are linked to poorer psychological well-being, increased disability, and accelerated cognitive decline [1, 11]. Our results indicate that depressive symptoms and informant-reported functional limitations—both potentially modifiable—are key predictors of less adaptive ageing attitudes. These findings support the value of early interventions targeting both psychological health and everyday function, using approaches such as cognitive-behavioural therapy, social prescribing, or functional reablement to foster more positive self-perceptions of ageing [56, 57].”
“Finally, the STAM tool—while a validated performance-based IADL assessment—may have been limited by ceiling effects in our high-functioning sample, potentially underestimating associations with ageing attitudes.”
Comment 6: Minor observation: 1. Basically, it is necessary to explore what is the unanswered research question in the literature that the authors intend to answer or at least what gap within this problem they want to fill.
Response 6: We have now revised the end of the Introduction to more clearly articulate the specific research gap our study addresses. While previous research has shown that both depressive symptoms and functional decline are associated with poorer ageing attitudes, there is limited understanding of how these predictors interact and which types of functional measures (i.e., performance-based vs. informant-reported) are most strongly associated with different domains of ageing attitudes over time.
Our study fills this gap by:
(1) incorporating both self- and informant-reported measures of IADL,
(2) assessing how these relate to three distinct subdomains of ageing attitudes (psychosocial loss, physical change, and psychological growth), and
(3) following very old adults (75+) longitudinally over a six-year period—an age group rarely the focus of SPA research despite being most vulnerable to internalised ageism and functional loss.
(See Introduction, lines 98 - 115.)
“Furthermore, despite growing evidence linking depressive symptoms and functional impairment to more negative attitudes toward ageing (ATA), key gaps remain. Most prior research has relied on self-reported functional measures and examined global ATA rather than specific domains (e.g., psychosocial loss, physical change, psychological growth). Additionally, the relative contributions of performance-based versus informant-reported functional measures to the development of ageing attitudes have not been compared. This is particularly important in very old adults (aged 75 and over), a population at increased risk for functional decline, depressive symptoms, and internalised ageism [28, 29], but often underrepresented in longitudinal research. The current study addresses these gaps by examining how depressive symptoms and both informant-reported and performance-based instrumental activities of daily living (IADLs) predict three distinct domains of ageing attitudes (physical change, psychosocial loss, and psychological growth) over a six-year period in a large cohort of community-dwelling older adults, aged 75 and above. Few studies have explored these relationships longitudinally or in very old adults—a group particularly vulnerable to both internalised stigma and adverse health outcomes. Drawing on stereotype embodiment theory [30], which posits that internalised ageing stereotypes impact health trajectories, we explore how psychological and functional experiences shape older adults’ perceptions of ageing.”
Reviewer 2 Report
Comments and Suggestions for Authors
The research topic concerning functional impairment, symptoms of depression and attitudes towards ageing in patients with dementia is indeed extremely relevant and important for clinical practice. In an ageing population where dementia is becoming more common, understanding the relationship between these aspects is crucial to improve the quality of life of patients.A large sample of patients was included in the study and long-term follow-up was conducted. Such studies are very valuable.
The authors carefully analysed the many factors affecting the patients and used adequate statistical methods to process the data.The authors also detailed the limitations of their study. Overall, I really enjoyed such a relevant paper with a unique methodology.
However, there are a few areas that could have been improved. For example, the inclusion of more extensive epidemiological data on the prevalence of anxiety-depressive conditions among older adults in the introduction could have highlighted the relevance of the topic and the importance of diagnosing these conditions in patients with dementia. It would also be useful to add at the end of the discussion strategies for treating depressive conditions in such patients and provide practical recommendations for clinicians.
Author Response
Comment 1: The research topic concerning functional impairment, symptoms of depression and attitudes towards ageing in patients with dementia is indeed extremely relevant and important for clinical practice. In an ageing population where dementia is becoming more common, understanding the relationship between these aspects is crucial to improve the quality of life of patients. A large sample of patients was included in the study and long-term follow-up was conducted. Such studies are very valuable.
The authors carefully analysed the many factors affecting the patients and used adequate statistical methods to process the data. The authors also detailed the limitations of their study. Overall, I really enjoyed such a relevant paper with a unique methodology.
Response 1: We thank the reviewer for taking the time to review our paper and providing their positive feedback.
Comment 2: However, there are a few areas that could have been improved. For example, the inclusion of more extensive epidemiological data on the prevalence of anxiety-depressive conditions among older adults in the introduction could have highlighted the relevance of the topic and the importance of diagnosing these conditions in patients with dementia.
Response 2: We agree with this and have added a sentence into the introduction to include epidemiological evidence of the prevalence of depression in the world’s ageing population to emphasise the relevance of our research topic.
(See Introduction, line 42-45.)
“This is all the more important given the high prevalence of mental health conditions in an ageing population, with depression affecting approximately 35.1% of older adults worldwide [7], and the number of people aged over 60 expected to reach 2.1 billion by 2050 [8].”
Comment 3: It would also be useful to add at the end of the discussion strategies for treating depressive conditions in such patients and provide practical recommendations for clinicians.
Response 3: We agree with this suggestion and have added a sentence to the discussion to reference clinical interventions that target depressive conditions and functional impairment, such as cognitive-behavioural therapy and functional reablement.
(See Discussion, line 400 – 403.)
“These findings support the value of early interventions targeting both psychological health and everyday function, using approaches such as cognitive-behavioural therapy, social prescribing, or functional reablement to foster more positive self-perceptions of ageing [56, 57].”
Reviewer 3 Report
Comments and Suggestions for Authors
This is well-written and interesting paper that examines longitudinally the impact of different parameters such as depressive symptoms and instrumental activities of daily living on attitudes toward ageing in a sample of Australian community-dwelling adults without dementia. The text is easy to follow and has a logical construction, the results and tables are presented in a clear and comprehensible manner and the study limitations are discussed in an analytic and explanatory way.
I have only two comments to make:
Materials and Methods
- Figure 1: At flowchart and in the last box, it is not clear if in the study sample are participating 25 patients with dementia out of 175 participants. This point needs clarification.
- I suggest the authors to add a title in the text for the statistical methods section.
Author Response
Comment 1: This is well-written and interesting paper that examines longitudinally the impact of different parameters such as depressive symptoms and instrumental activities of daily living on attitudes toward ageing in a sample of Australian community-dwelling adults without dementia. The text is easy to follow and has a logical construction, the results and tables are presented in a clear and comprehensible manner and the study limitations are discussed in an analytic and explanatory way. I have only two comments to make:
Response 1: We thank the reviewer for providing such positive feedback and helpful suggestions.
Comment 2: 1. Figure 1: At flowchart and in the last box, it is not clear if in the study sample are participating 25 patients with dementia out of 175 participants. This point needs clarification.
Response 2: We agree with the reviewer. The 25 participants were those within the included sample who had progressed to dementia during the 6 years of follow up (wave 4 to wave 7). We have edited Figure 1 to clarify this.
(See Figure 1, Materials and Methods, Line 148.)
“Incident dementia (within the included): 25”
Comment 3: 2. I suggest the authors to add a title in the text for the statistical methods section.
Response 3: We agree with this reviewer. A title for Statistical Analysis was added for ease of navigation and signposting.
(See Materials and Methods, line 210.)